# Thiacalixarenes with Sulfur Functionalities at Lower Rim: Heavy Metal Ion Binding in Solution and 2D-Confined Space

**DOI:** 10.3390/ijms23042341

**Published:** 2022-02-20

**Authors:** Anton Muravev, Ayrat Yakupov, Tatiana Gerasimova, Daut Islamov, Vladimir Lazarenko, Alexander Shokurov, Alexander Ovsyannikov, Pavel Dorovatovskii, Yan Zubavichus, Alexander Naumkin, Sofiya Selektor, Svetlana Solovieva, Igor Antipin

**Affiliations:** 1Arbuzov Institute of Organic and Physical Chemistry, FRC Kazan Scientific Center, Russian Academy of Sciences, Arbuzov Str. 8, 420088 Kazan, Russia; gryaznovat@iopc.ru (T.G.); daut1989@mail.ru (D.I.); osaalex2007@rambler.ru (A.O.); svsol@iopc.ru (S.S.); 2Butlerov Institute of Chemistry, Kazan Federal University, Kremlevskaya Str. 18, 420008 Kazan, Russia; yakupov.airat@bk.ru (A.Y.); iantipin54@yandex.ru (I.A.); 3National Research Centre “Kurchatov Institute”, Kurchatov Square 1, 123182 Moscow, Russia; vladimir.a.lazarenko@gmail.com (V.L.); paulgemini@mail.ru (P.D.); 4Frumkin Institute of Physical Chemistry and Electrochemistry, Russian Academy of Sciences, Leninsky Pr. 31, 119071 Moscow, Russia; dr.shokurov@gmail.com (A.S.); naumkin@ineos.ac.ru (A.N.); sofs@list.ru (S.S.); 5Synchrotron Radiation Facility SKIF, Boreskov Institute of Catalysis, Siberian Branch, Russian Academy of Sciences, Nikolskii Pr. 1, 630559 Koltsovo, Russia; yzubav@gmail.com; 6Nesmeyanov Institute of Organoelement Compounds, Russian Academy of Sciences, Vavilov Str. 28, 119334 Moscow, Russia

**Keywords:** thiacalix[4]arenes, Langmuir monolayers, thiacrown-ethers, X-ray photoelectron spectroscopy, Ag^+^ and Hg^2+^ complexes

## Abstract

Sulfur-containing groups preorganized on macrocyclic scaffolds are well suited for liquid-phase complexation of soft metal ions; however, their binding potential was not extensively studied at the air–water interface, and the effect of thioether topology on metal ion binding mechanisms under various conditions was not considered. Herein, we report the interface receptor characteristics of topologically varied thiacalixarene thioethers (linear bis-(methylthio)ethoxy derivative **L^2^**, O_2_S_2_-thiacrown-ether **L^3^**, and O_2_S_2_-bridged thiacalixtube **L^4^**). The study was conducted in bulk liquid phase and Langmuir monolayers. For all compounds, the highest liquid-phase extraction selectivity was revealed for Ag^+^ and Hg^2+^ ions vs. other soft metal ions. In thioether **L^2^** and thiacalixtube **L^4^**, metal ion binding was evidenced by a blue shift of the band at 303 nm (for Ag^+^ species) and the appearance of ligand-to-metal charge transfer bands at 330–340 nm (for Hg^2+^ species). Theoretical calculations for thioether **L^2^** and its Ag and Hg complexes are consistent with experimental data of UV/Vis, nuclear magnetic resonance (NMR) spectroscopy, and single-crystal X-ray diffractometry of Ag–thioether **L^2^** complexes and Hg–thiacalixtube **L^4^** complex for the case of coordination around the metal center involving two alkyl sulfide groups (Hg^2+^) or sulfur atoms on the lower rim and bridging unit (Ag^+^). In thiacrown **L^3^**, Ag and Hg binding by alkyl sulfide groups was suggested from changes in NMR spectra upon the addition of corresponding salts. In spite of the low ability of the thioethers to form stable Langmuir monolayers on deionized water, one might argue that the monolayers significantly expand in the presence of Hg salts in the water subphase. Hg^2+^ ion uptake by the Langmuir–Blodgett (LB) films of ligand **L^3^** was proved by X-ray photoelectron spectroscopy (XPS). Together, these results demonstrate the potential of sulfide groups on the calixarene platform as receptor unit towards Hg^2+^ ions, which could be useful in the development of Hg^2+^-selective water purification systems or thin-film sensor devices.

## 1. Introduction

Control of supramolecular assembly, in addition to known synthetic strategies toward diverse molecular architectures, shapes many characteristics of emerging materials. Among these architectures, basket-like calixarene macrocycles are particularly attractive due to their predictable conformational behavior and ability to pre-organize a variety of functional groups through stereo- and regioselective modification [1]. As a result, not only superior characteristics of materials but also fundamental aspects of the structure–property relationship could be characterized, which was highlighted in recent reviews [2,3].

Sulfur functionalities were among the first groups tested in calixarenes due to their binding ability toward toxic metal ions [4,5], and their analytical detection is still relevant, as exemplified by the work on chromogenic recognition of Cu^2+^ ions by sulfonated thiacalix[4]arene [6]. At present, the scope of application of sulfur functional groups on calixarene scaffolds is quite diverse, and some representative examples are given in Figure 1. Porous polymers based on Sonogashira-coupled pyrene–calix[4]thiacrown conjugates **I** displayed record-high Hg^2+^ absorption efficiency [7]. In our group, enzyme-binding *1,3-alternate* thiacalix[4]crown-ether **II** appended with thioester units was transferred onto a gold substrate via Langmuir–Schaefer film formation [8]. Sulfur units in heterotopic thiacalix[4]arene **III** performed an anchoring function to immobilize the compound on Au nanoparticles through dithiolane fragments and performed receptor function toward ultralow quantities of Ag^+^ ions via O_2_S_2_O-crown-ether [9]. A good example of a solid-state metal-coordination framework was obtained for the complex of CuI and bisthiacrown receptor **IV**, which displayed photoluminescence switching [10].

To date, the receptor ability of hydrophobic thioether units remains relatively unexplored at the air–water interface, where Langmuir monolayers are produced, and only a few compounds have been reported [11,12]. Given the outstanding binding characteristics of thioether units preorganized on a calixarene macrocycle, as well as unique coordination features with soft metal ions, understanding the interaction of these ligands with metal ions in confined space is highly demanding for sensor design and related applications.

In this work, three receptors with sulfur-containing units with a different preorganization degree, from linear thioether **L^2^** to capped thiacrown **L^3^** and thiacalixtube **L^4^** (Figure 1), were synthesized from thiacalix[4]arene **L^1^**. Their complexation ability toward heavy metal ions was evaluated in solution and Langmuir monolayers in order to discover if there are effective thioether–metal interactions at the air–water interface and if receptor topology alters the metal ion binding mechanism or selectivity.

## 2. Results

### 2.1. Synthesis and Characterization of Ligands

Thiacalix[4]arenes can undergo selective etherification at hydroxyl groups through a Mitsunobu reaction with primary alcohols in the PPh_3_/EtOOC–N=N–COOEt system, which affords distally disubstituted ether derivatives [13]. Using this synthetic approach, thiacalix[4]arene **L^1^** [14] was functionalized at the lower rim by appropriate alcohols, as reported previously, and converted into thioether **L^2^** [15] and thiacalixtube **L^4^** [16] (Figure 1). It is obvious that *1,3-alternate* thiacrown **L^3^** could hardly be accessed through the sequential addition of 3,6-dithia-1,8-octanediol and dodecanol due to favorable [2 + 2]-macrocyclization of thiacalixarene **L^1^** into thiacalixtube **L^4^** during the first stage. Therefore, an inverse alkylation sequence via the previously synthesized dodecyl intermediate **L’^1^** [17] was employed (with the reaction of thiacalixarene **L^1^** with dodecanol). Late-stage capping of two OH groups of the dodecyl intermediate **L’^1^** with diol afforded exclusively the product of [1 + 1]-macrocyclization (a peak of fragmented molecular ion at *m*/*z* 1193.9 [M − S + Na]^+^).

Compound **L^3^** was fully characterized by ^1^H/^13^C NMR (Appendix A) and infrared (IR) spectroscopy and matrix-assisted laser desorption-ionization (MALDI) mass spectrometry. Spacing between NMR signals of the protons of the aryl ring is a well-known criterion of conformation assignment in calixarene derivatives. It could distinguish *cone* and *1,3-alternate* stereoisomeric forms possessing identical splitting patterns of NMR signals. In the former case, two diagonal aryl rings impose a shielding ring current effect on neighboring aryl rings, resulting in their significant upfield shift and a large spacing between these signals [18,19]. The ring current effect is impossible in the *1,3-alternate* stereoisomeric form, and the aryl signals are located close to each other. Thus, the existence of compound **L^3^** in the *1,3-alternate* stereoisomeric form was confirmed by closely spaced NMR signals of aryl protons at 7.41 and 7.35 ppm and *tert*-butyl protons at 1.31 and 1.29 ppm.

### 2.2. Complexation of Metal Ions at Liquid–Liquid Interface

Initial assessment of non-competitive binding of heavy metal ions (Ag^+^, Cd^2+^, Cu^2+^, Hg^2+^, Pb^2+^, and Zn^2+^) in the form of picrates by the thioethers **L^2^**–**L^4^** was carried out by liquid-phase extraction. At this stage, there were no significant differences between the thioethers and Ag^+^ and Hg^2+^ extractability was always highest among the metal ions (Figure 2a). A decrease in the degree of extraction of *1,3-alternate* thiacrown **L^3^** (44–67%) as compared to *cone* thioethers **L^2^** and **L^4^** (higher than 90%) could be caused by the steric hindrance of the thiacrown receptor unit by neighboring aryl rings. The extractability of Cd^2+^, Cu^2+^, Pb^2+^, and Zn^2+^ picrates by the ligands **L^2^**–**L^4^** was less than 5%.

To evaluate the binding strength of the calixarene ligands toward metal ions and establish the stoichiometry of the extracted metal complexes, the extraction constants were calculated from the degrees of extraction of metal picrate salts (*c* = 5 × 10^–5^ M in the case of Hg^2+^ and *c* = 1 × 10^–4^ M in the case of Ag^+^) at varying concentrations of ligands (*c* = 10^–6^–10^–4^ M). Linear fit of the scatter plot of log*D*_Ag_ (*D* is the distribution ratio of metal ions between organic and aqueous phases) against log*c* of the thioethers gave the slope close to unity, which indicates that silver picrate is extracted as the 1:1 complex (Appendix A). The concentration range, in which there was a linear dependence of log(*D*_Hg_/(1–α)) (α is the degree of extraction (α = %*E*/100)) against log*c*, was much narrower than that of Ag^+^ and a rough assumption of the predominant ligand-to-metal stoichiometry gives 1:3 (**L^3^**), 1:2 (**L^2^** and **L^3^**), and 1:1 (**L^4^**) complexes (Appendix A). Comparison of the extraction constants of silver complexes of the thioethers (Table 1) agrees with the extraction percentage values of silver picrates and shows that the linear thioether and thiacalixtube extract Ag^+^ better than thiacrown **L^3^**. Different stoichiometry of the extracted Hg species does not allow for a direct comparison of the extraction effectiveness by the thioether ligands.

The nonlinear dependence of log*D*_Ag_ and log(*D*_Hg_/(1–α)) vs. log*c* at high concentrations of ligand suggests the effect of aggregation of ligand and metal picrate apart from high ligand-to-metal ratio within the complexes. It was previously hypothesized that Ag^+^ extraction facilitates aggregate formation in the organic phase in the case of thiacalixtube **L^4^** due to the higher thermodynamic stability of aggregates as compared to monomer complexes (hydrodynamic particle diameter was 244 ± 4 nm according to dynamic light scattering (DLS) data) [16]. DLS particle size distribution analysis in the organic phase after heavy metal ion extraction by the ligands (*c*_L_ = 4 × 10^−4^ M in CH_2_Cl_2_) revealed no temporally stable aggregates in the case of thiacrown **L^3^**, whereas stable spherical aggregates were detected after extraction of Hg^2+^ (*d* = 437 ± 12 nm) and Ag^+^ ions (*d* = 347 ± 12 nm) by the thioether **L^2^** (Figure 2b). To verify that aggregate formation is related to the complexation of metal ions in the form of picrate salt, a number of reference experiments were carried out for Hg^2+^ ions involving the mixing of (1) compound **L^2^** in CH_2_Cl_2_ with a 2 × 10^−4^ M aqueous solution of picric acid, (2) compound **L^2^** in CH_2_Cl_2_ with an aqueous solution of Hg(NO_3_)_2_, and (3) blank CH_2_Cl_2_ with Hg(NO_3_)_2_ dissolved in a 2 × 10^−4^ M aqueous solution of picric acid. When picric acid was added, it was dissolved in tris hydrochloride buffer up to pH 5.8 (0.05 M) to avoid partial hydrolysis of mercury picrate salt and its possible interference into DLS measurements. Only turbid suspensions with microparticles were detected in the organic phase, which presumably corresponds to micelle-stabilized water-in-oil microemulsions (Appendix A). The absence of nanosized aggregates in these reference experiments suggests that aggregates detected in the organic phase after extraction of mercury picrate are associated with the complexation with HgPic_2_ salt. When lower concentrations of the ligands were employed (within the linear range log*D*_Ag_ and log(*D*_Hg_/(1–α)) vs. log*c* plots), only particles with a size of less than 10 nm were detected (corresponding to individual complexes).

### 2.3. Solid-State Structures of Ag^+^ and Hg^2+^ Complexes

The preferred binding mode of Ag^+^ and Hg^2+^ ions was further analyzed by X-ray diffractometry (crystallographic parameters are given in Appendix A). The crystals grown via diffusion between thiacalixtube **L^4^** in CHCl_3_ and HgCl_2_ in MeOH feature a triclinic *P*-1 space group in the unit cell and were identified as monomeric binuclear complex [Hg_2_**L^4^**Cl_4_] (Figure 3). Due to the *distorted cone* conformation of the thiacalixarene backbone with parallel orientation of ether-containing aryl fragments on the lower rim in the thiacalixtube, the distance between S atoms in a linker is shorter (3.56 Å) than that between S atoms of different linkers (5.59 Å) or between aryl and alkyl sulfur atoms (4.12–4.70 Å); this difference explains the convergent exo-coordination mode of Hg^2+^ ions involving two sulfur donors on the lower rim. The other two coordination sites occupied by Cl^–^ ions result in a distorted tetrahedral geometry of the Hg atom (Appendix A).

The Ag^+^ ions were expected to coordinate both aryl and alkyl sulfide atoms, and one can assume different topologies of the complexes depending on the metal-to-ligand ratio. Indeed, a dinuclear cage-like [Ag_2_**L^2^**_2_](ClO_4_)_2_ complex was formed at an equimolar component ratio (Figure 4), with the triclinic *P*-1 space group similar to that of a previously reported thiacalixcrown–Ag self-assembled dimer [20]. In contrast to the mentioned work, each Ag^+^ ion adopts a distorted trigonal coordination in the complex with aryl and alkyl sulfide units from one molecule and an alkyl sulfide unit from another molecule, forming an S_3_ binding motif (Appendix A). The *distorted cone* conformation of calixarene macrocycle was again preserved.

On addition of a 10-fold excess of AgClO_4_, a zig-zag 1D polymeric array with the formula [Ag_2_**L^2^**(ClO_4_)_2_]*_n_* is formed, which crystallizes in the orthorhombic *Pbca* space group (Figure 5). In the core coordination unit of the 1D chain, the Ag^+^ ion adopts a distorted tetrahedral geometry with two oxygen atoms of two linker ClO_4_^–^ ions and two sulfur atoms from the bridging unit and lower rim (Appendix A). Similar uncommon behavior of the ClO_4_^–^ anion as a linker in the dimensional framework has been recently documented for the 2D network of the Ag complex of thiacalix[4]biscrown-ether [21]. Interestingly, thiacalix[4]arene adopts a more symmetrical *pinched cone* conformation in this complex, which is normally observed in methylene-bridged calix[4]arenes. As a result, two H-bonds with different O atoms of ether fragments are formed.

### 2.4. Metal Ion Binding in Solution

Spectrophotometric titration was employed to gain further evidence of metal ion binding in solution. Figure 6 demonstrates the evolution of UV/Vis spectra on the addition of AgClO_4_ and Hg(ClO_4_)_2_ to thioether **L^2^** in 10:1 CH_2_Cl_2_–MeOH solution (this system was chosen due to its dissolving ability toward all components). Titration of the ligand with AgClO_4_ resulted in a 5-nm blue shift of the band at 303 nm from 0 to 1.5 eq. of metal salt and did not change upon its further addition (Figure 6a). The addition of more than 2 eq. of AgClO_4_ leads to the appearance of a weak absorption at 336 nm. This weak absorption at 336 nm was also observed when AgOTf was employed instead of AgClO_4_. Similarly, titration with Hg(ClO_4_)_2_ gave rise to a shoulder centered at 330 nm, an intensity of which reached a plateau at 3 eq. of the metal salt (Figure 6b). Thiacalixtube **L^4^** displays essentially the same spectroscopic response toward metal ions as ligand **L^2^** (Appendix A).

UV/Vis spectra of thiacrown **L^3^** weakly changed upon titration with AgClO_4_ and Hg(ClO_4_)_2_ and no clear information on the binding mechanism of these ions can be derived.

Upon addition of Pb^2+^, Zn^2+^, and Cd^2+^ ions to the ligands or the mixtures of ligands with Ag^+^ or Hg^2+^ ions, there was a non-specific increase in absorbance at 240–275 nm due to the absorbance of metal perchlorate. Thus, the thioethers selectively interact with Ag^+^ and Hg^2+^ ions in solution, and this process is not interfered with by other soft metal ions. However, the stability constants are quite low since there are continuous spectral changes even at a large excess of metal salts. To evaluate the interference of Ag^+^ and Hg^2+^ ions under competitive titration conditions, 0.1 to 100 eq. of AgClO_4_ and Hg(ClO_4_)_2_ were added to the **L^2^**–Hg(ClO_4_)_2_ or **L^2^**–AgOTf system (L:M ratio is 1:10). In both cases, the spectral changes intrinsic for the added metal salt were observed (appearance of the band at 336 nm in the case of Hg^2+^ and a blue shift of the band at 303 nm in the case of Ag^+^), whereas the features of the initial **L^2^**–metal salt system remained intact even at 100 eq. of the added metal salt.

UV/Vis spectra highlight not only a selective response of thioethers **L^2^** and **L^4^** to Ag^+^ and Hg^2+^ ions, but also its different origin depending on metal ion and ligand topology. Firstly, regarding Ag^+^ ions, there are OS/O_2_S binding motifs in thioethers **L^2^** and **L^4^** involving epithio groups between aryl rings, which are well suited for Ag^+^ ions [20,22,23]. Thus, cooperative Ag^+^ binding by OS/O_2_S motifs and alkyl sulfide units in compounds **L^2^** and **L^4^** could result in a hypsochromic shift of absorption at 303 nm and the appearance of a weak band at 336 nm. Analysis of density functional theory (DFT) calculated UV/Vis spectra of quantum-chemically optimized model complexes **L^2^-Ag-a**, **L^2^-Ag-b**, **L^2^-Ag-c**, and monomer unit **L^2^-Ag-XRay** with two Ag^+^ ions from X-ray analysis (Figure 5) with different coordination environments (Figure 7) showed a similar blue shift of the longest wavelength for model complexes **L^2^-Ag-a**, **L^2^-Ag-b**, and **L^2^-Ag-XRay** compared to ligand **L^2^** spectrum agrees with experimental observations (Figure 6). However, neither model predicts a shoulder at 336 nm. According to the literature, the formation of counterion-linked Ag coordination polymers (Cl^–^ and even weakly coordinating BF_4_^–^) can lead to a red shift of the longest wavelength absorption bands related to metal-to-ligand charge transfer [24,25]. Thus, one can assume that a large excess of AgClO_4_ promotes the formation of 1D coordination polymer [Ag_2_**L^2^**(ClO_4_)_2_]*_n_* (Figure 5), which absorbs at higher wavelengths as compared to monomeric units.

Secondly, regarding Hg^2+^ ions, there is less information on the binding mechanism of Hg^2+^ ions in sulfur-containing calixarenes, with a direct observation of coordination of Hg^2+^ ions with lower-rim sulfide units [26] and proposed binding with sulfide bridges [27]. In this work, only charge-transfer bands at 330 nm for thioethers **L^2^** and **L^4^** were reliably identified upon titration of Hg^2+^; however, the origin of the electron-donating group is unclear. To shed light on the possible structure of the **L^2^-Hg** complex, three model complexes were optimized quantum-chemically: (i) **L^2^-Hg-a** with Hg^2+^ ion bound to two alkyl sulfide units in an intramolecular fashion and two ClO_4_^–^ anions, similar to [26], (ii) **L^2^-Hg-b** with Hg^2+^ ion bound to both aryl sulfide and alkyl sulfide groups, and (iii) **L^2^-Hg-c** with Hg^2+^ ion inside the macrocycle cavity (Figure 7). Analysis of density functional theory (DFT) calculated UV/Vis spectra of the model complexes and ligand **L^2^** (Figure 7) showed the best agreement of the theoretical spectrum of **L^2^-Hg-a** with the experimental curve (Figure 6). Analysis of frontier molecular orbitals contributing to the band maxima at 330 nm in the spectrum of the **L^2^-Hg-a** complex suggests that it has a ligand-to-metal charge transfer origin (Appendix A), whereas the longest wavelength absorption in the spectrum of ligand **L^2^** is largely contributed by the orbitals localized on benzene rings (Appendix A). The second and third models of ion binding were unfeasible due to the large difference between the predicted absorption maxima in the range of 390–600 nm and those in the experimental spectra.

Further support of the mechanism of metal ion binding was provided from comparison of ^1^H NMR spectra of the thioethers **L^2^** and **L^3^** with and without AgClO_4_ and Hg(ClO_4_)_2_. In analogy to UV/Vis titration experiments, a 10:1 CDCl_3_–CD_3_OD solvent was first chosen; however, the metal salts did not dissolve in this system due to the higher concentration (*c* = 2 × 10^−2^ M); therefore, CD_3_OD was replaced by CD_3_CN. Under these conditions, a true (non-colloidal) solution was formed (also verified by DLS data).

The rather low change of proton resonances in OH groups and aromatic ring and *t*-Bu group (Δ*δ* < 0.10 ppm) suggest retention of the calixarene conformation upon complexation (Figure 8), which agrees with the retention of stereoisomeric form in crystal state at low equivalents of metal ion (Figure 3 and Figure 4). On the contrary, large downfield chemical shifts of the protons of CH_2_SMe group in compound **L^2^**–Ag^+^ system (Δ*δ*(CH_2_) ≈ 0.16–0.91 ppm, Δ*δ*(CH_3_) ≈ 0.26–0.90 ppm) and thiacrown in the ligand **L^3^**–Ag^+^ system (Δ*δ* = 0.19–0.69 ppm) clearly indicate the involvement of the lower-rim thioether group in the binding of Ag^+^ species. A less apparent shift of the signal of OCH_2_ protons in the thiacrown ring in compound **L^3^** (0.19 ppm) could also be induced by a conformational change of the crown-ether macrocycle. Therefore, encapsulation or exocyclic coordination of the Ag^+^ ion by the thiacrown macrocycle, which was evidenced in solid state for thiacalix[4]thiacrown-ethers [28], are equally possible from solution data.

Large peak broadening on addition of 2 and 4 eq. of Hg(ClO_4_)_2_ to thioether **L^2^** and splitting of the triplets of CH_2_ groups into two broad signals should also be noted (4.72 and 4.21 ppm for OCH_2_ and 4.06 and 3.67 ppm for SCH_2_). Such peak broadening indicates fast exchange between complexed and uncomplexed species relative to the NMR time scale. The observed splitting pattern suggests either the presence of several different complexes or the low symmetry of the complex. The latter event is presumably not related to poor solvation of the perchlorate anion in low-donor solvents, as postulated in [29], because no peak splitting was observed in the case of AgClO_4_ (Figure 8). Doubling of the signals of aryl protons at 7.04 ppm and *tert*-butyl protons at 1.30 ppm indicates a growing discrimination within aryl and *tert*-butyl groups upon complexation and again supports the hypothesis of low symmetry of the complex. Similarly, the signals of OCH_2_CH_2_S fragment doubled upon the addition of 1 eq. Hg^2+^ to thiacrown **L^3^** and there was a down-field shift of the CH_2_ singlet (Δ*δ* ≈ 0.68 ppm), whereas further addition of Hg^2+^ salt (2 eq.) resulted in large peak broadening and the spectrum could not be analyzed. The above chemical shifts appear not to be induced by the change of ionic strength of solution or the presence of water molecules in metal perchlorate salts because Cd(ClO_4_)_2_·xH_2_O (x ≈ 6) did not alter the NMR spectra of the ligands.

Solution study unequivocally indicates that thioether units on the lower rim of the calixarene macrocycle are involved in Ag^+^ and Hg^2+^ binding regardless of the topology of the ligands **L^2^**–**L^4^**, and the results are consistent with solid-state structural features. The ion binding motif in thioethers is summarized in Table 2.

### 2.5. Aggregation and Interactions with Ions at Air–Water Interface

Thioethers are not conventional monolayer-forming compounds due to their low hydrophilicity. Consequently, attachment of such units to the calixarene backbone makes it hard to predict molecular orientation at the air–water interface, and the metal binding ability of thioethers under these conditions is a priori unclear. On the one hand, metal ions render thioether hydrophilic upon complexation, and true Langmuir monolayers can be formed. On the other hand, sulfur atoms of the thiacalix[4]arene macrocycle rim were ineffective in Ag^+^ binding at the air–water interface [30], which suggests a high barrier to transfer these moieties from air to water.

To form monolayers, 5 × 10^–5^ M solutions of thioethers **L^2^**–**L^4^** in CHCl_3_ were spread over the water subphase to give the mean molecular area larger than that at parallel orientation of C_2v_ axes of molecules relative to the water subphase ($A_{∥}$). Monolayers were characterized by (surface pressure *π*)–(mean molecular area *A*) isotherms recorded upon monolayer compression (Figure 9).

In the case of *cone* stereoisomers **L^2^** and **L^4^**, take-off areas *A*_0_ in *π*–*A* isotherms (176 and 277 Å^2^) are higher than $A_{∥}$ values of thioethers corresponding to the parallel orientation of C_2v_ axes of molecules relative to the water subphase. As expected, thioether **L^2^** does not form well-organized monolayers, and the critical pressure (pressure corresponding to the kink on isotherm at low areas per molecule) at ca. 7 mN/m (at *A* ≈ 130 Å^2^ corresponding to perpendicular orientation of C_2v_ axes (*A*_⊥_)) indicates weak interactions between water molecules and thiacalixarene. A sufficiently high critical pressure was recorded for the monolayer of thiacalixtube **L^4^**, 24 mN/m. In spite of rigid monolayer formation in the case of ligand **L^4^**, there are no strong irreversible interactions between molecules in the monolayer, which was shown by the coincidence of compression and expansion cycles of *π*–*A* isotherms (Appendix A).

In the case of *1,3-alternate* thiacrown **L^3^**, surface pressure increased over compression at the spreading area of 300–500 Å^2^ from 161 ± 5 Å^2^ up to the mean molecular areas much lower than the macrocycle cavity size. Reproducibility of the *π*–*A* isotherm of compound **L^3^**, the absence of hysteresis on it upon cycling, and similarity of UV/Vis spectra in solution and at the air–water interface (Appendix A) indicate not only the absence of aggregation but also feeble interaction between molecules in the monolayer. This behavior can be explained by the reversible extrusion of some molecules from the monolayer upon compression. At a higher spreading area of ligand **L^3^** (600 Å^2^), its isotherm was similar to that of ligand **L^4^**, with *A*_0_ = 307 ± 9 Å^2^.

The effect of Ag^+^, Hg^2+^, Cu^2+^, and Pb^2+^ ions as nitrate salts on monolayer characteristics was further analyzed (Figure 10, Appendix A). For compounds **L^2^** and **L^3^**, surface potential (SPOT) Δ*V* increased in the presence of Ag^+^ and Hg^2+^ salts due to the negative contribution of the ions of salt under the monolayer into the double-layer potential according to the Demchak–Fort model (Appendix A). This fact suggests incorporation of these ions into the monolayer and, consequently, a positive contribution into the dipole potential. Conversely, maximum SPOT values changed marginally in the case of Cu^2+^ and Pb^2+^ ions. Figure 10 shows that the slope of *π*–*A* curves does not change significantly in the presence of most metal salts in the water subphase, which indicates the retention of the liquid-expanded state of the monolayers. In the case of linear thioether **L^2^**, addition of Cu^2+^ and Pb^2+^ ions to the water subphase altered the take-off area of the ligand **L^2^** monolayer within 18 Å^2^, which is close to the measurement error (5 Å^2^) (Appendix A). When there are Ag^+^ ions in the water subphase, the take-off area of the thioether **L^2^** monolayer also remains nearly intact, yet its critical pressure increases from 7 to 25 mN/m (Figure 11). This result clearly indicates monolayer interaction with these metal ions and seems to be caused by the involvement of the alkyl thioether chain in ion binding. The largest monolayer expansion in the case of Hg^2+^ ions indicates a greater response of thioether **L^2^** monolayer toward this ion, presumably due to larger electrostatic repulsion as compared to Ag^+^ ions.

Similarly, the most significant shift of *π*–*A* isotherms to larger areas in the case of compound **L^3^** was recorded for Hg(NO_3_)_2_, whereas AgNO_3_ did not significantly change the take-off area.

In contrast to the former two thioethers, the characteristics of the thiacalixtube **L^4^** monolayer did not alter upon variation of the water subphase composition with the introduction of the metal nitrates under study, suggesting no binding at the air–water interface. There are two possible reasons for such an effect: (1) bridging sulfur atoms between aryl rings and on the lower rim are not immersed into the water subphase and (2) the receptor ability of sulfur functionalities in thiacalixtube topology is insufficient to desolvate the metal cation in the form of nitrate salt. It is known that nitrate anions are considered kosmotropic (contributing to the structure of water–water interactions), and replacement of these anions by chaotropic (structure-breaking) anions would decrease the solvation of the cation [31]. Therefore, a number of chaotropic anions (Pic, ClO_4_, OTf, and OTs) in the form of silver salts were tested (Figure 10). In all these cases, the thiacalix[4]tube monolayer significantly expanded, indicating an interaction with the metal salt. The following order of monolayer expansion was obtained: NO_3_^–^ < Pic^–^ ≈ ClO_4_^–^ < OTs^–^ < Otf^–^. To verify that cation selectivity is preserved, Hg(ClO_4_)_2_ was added to the water subphase, which resulted in the largest monolayer expansion to the take-off area of ca. 500 Å^2^. Therefore, ligand **L^4^** can also bind Hg^2+^ and Ag^+^ ions at the air-water interface.

To prove the incorporation of Hg^2+^ ions into the monolayers of thioethers at the air–water interface, a surface-sensitive XPS was employed. For this purpose, the example monolayer of thiacrown **L^3^** formed on the surface of 10^−4^ M aqueous solution of Hg(ClO_4_)_2_ (a lower concentration was used to avoid nonspecific adsorption of Hg^2+^ ions on the monolayer) was transferred onto the quartz substrate through the vertical dipping LB technique with a transfer ratio of 0.91. The resulting spectra of the quartz plate with the Langmuir–Blodgett film were recorded (full XPS spectra are given in Appendix A). Due to the overlap of Hg4f and Si2p regions and a much lower Hg content on quartz compared to silicon within the depth of XPS analysis, both Si2s and Si2p regions of XPS spectra were analyzed (Appendix A). The irrefutable proof of Hg presence at the quartz surface was gained from deconvolution of the Si2p region into components (Figure 11). In addition to the peaks with binding energies of 99.34 and 99.95 (state Si^0^) and 100.54 (Si^+^), 101.64 (Si^2+^), and 103.65 eV (Si^4+^), two extra peaks attributed to well-separated spin-orbit components of Hg4f were identified, with bond energies of 105.42 eV (Hg4f_5/2_) and 101.16 (Hg4f_7/2_) and intensity ratio of 0.75. Detailed characteristics of the peaks are given in Appendix A. It could be speculated that the SiO_2_ surface not covered by the monolayer could carry water and interact with the studied ions due to its negative charge at a neutral pH. Therefore, as a negative control, the surface of the bare quartz was immersed into 10^−4^ M Hg(ClO_4_)_2_ and, then, dried in the air for 24 h. In this case, no XPS peaks of Hg^2+^ ions were detected (Appendix A).

In summary, monolayers of all three thioethers **L^2^**–**L^4^** with different degrees of pre-organization of sulfur functional groups displayed similar selectivity toward heavy metal ions. In contrast to the CH_2_Cl_2_–water system, where for all thioethers, Ag^+^ and Hg^2+^ extractability was largest, the extent of ligand **L^2^**–**L^4^** monolayer expansion at the air–water interface was highest in the case of Hg^2+^ salt. Regarding the demand to employ chaotropic salts to evidence ion–monolayer interactions in the case of thiacalixtube **L^4^**, linear thioether **L^2^**, and thiacrown **L^3^** show higher potential as receptor units towards Ag^+^ and Hg^2+^ ions in ultrathin films. Thus, preorganization of sulfur functionalities from the linear chain in compound **L^2^** to the crown-ether macrocycle **L^3^** and tube in compound **L^4^** seems not to promote the receptor ability of the ligand at the air–water interface. This finding is opposite to the trend recently determined by us for oxyethylene derivatives on the lower rim of thiacalixarene derivatives [30]. In that work, the oxacrown-5 macrocycle on the *1,3-alternate* thiacalixarene scaffold demonstrated Ca^2+^ selectivity in Langmuir monolayers expressed by the degree of monolayer expansion, whereas monolayer characteristics of linear oligooxyethylene counterpart did not change with the addition of Ca^2+^ ions to the water subphase. This difference can be rationalized by different coordination modes of metal cations by the literature oxacrown ligands and thioether receptors studied in this work: exocyclic coordination of sulfur atoms, at which linear thioethers are equally well suited as cyclic counterparts (this study), and endocyclic coordination involving oxygen atoms, at which cyclic compounds provide gain in entropy as compared to podands due to pre-organization (oxyethylene derivatives in [30]).

The O_2_S_2_ motif in ligands **L^2^**–**L^4^** displays an analogous selectivity pattern among heavy metal ions (towards Ag^+^ and Hg^2+^ ions) as compared to the tetra-functionalized calixarene thioethers reported in the literature [5]. Linear disubstituted calixarene thioethers were only selective towards Ag^+^ ions according to extraction data [32] and Hg^2+^ ion selectivity arises only with a transition to calix[4]thiacrown-ether topology [33]. In the latter case, however, there is a rather high extractability of Cu^2+^ and Pb^2+^ ions. Provided that the synthetic route toward linear thioether **L^2^** is more facile than those toward calix[4]thiacrown-ethers and no interaction was recorded between the thioethers and Cu^2+^, Zn^2+^, Pb^2+^, and Cd^2+^ ions neither in solution state nor in confined space, selective Hg^2+^ absorption studies are particularly attractive for the synthesized compounds and better Hg^2+^ ion sensing or uptake could be expected.

## 3. Materials and Methods

Organic solvents were purified by known procedures [34]; the reagents, picric acid, tris hydrochloride, and metal salts (AgNO_3_, AgClO_4_·H_2_O, Cd(NO_3_)_2_·4H_2_O, Cd(ClO_4_)_2_·xH_2_O (x≈6), Cu(NO_3_)_2_·3H_2_O, Cu(ClO_4_)_2_·6H_2_O, Hg(NO_3_)_2_·H_2_O, Hg(ClO_4_)_2_·xH_2_O (x = 3–5), Pb(NO_3_)_2_, Pb(ClO_4_)_2_·xH_2_O (x ≈ 6), Zn(NO_3_)_2_·6H_2_O, and Zn(ClO_4_)_2_·6H_2_O were used as received. Water was deionized on an Adrona Crystal purification system up to *σ* = 0.055 µS/cm.

Lower-rim distal functionalization of thiacalix[4]arene **L^1^** [14] was realized in one stage by a Mitsunobu reaction in the PPh_3_/EtOOC–N=N–COOEt system using appropriate alcohols, as reported for thioether **L^2^** [15] and thiacalixtube **L^4^** [16] (Figure 1). The synthesis of thiacrown **L^3^** is described below.

NMR spectra were recorded at 30 °C on a Bruker AVANCE-400 and 600 MHz (400.05/600.13 MHz for ^1^H NMR and 100.6 MHz for ^13^C NMR) spectrometer. Chemical shifts are reported in the δ (ppm) scale relative to the ^1^H (7.26 ppm) and ^13^C (77.2 ppm) signals of CDCl_3_, and coupling constants are denoted in Hz. During NMR titration, trace solvent peaks were detected at 5.26 (CH_2_Cl_2_) and 1.92 ppm (CHD_2_CN). The chemical shift of water from metal perchlorate hydrate varied depending on the system (2.33 ppm (**L^2^**–Ag^+^), 2.21 ppm (**L^3^**–Ag^+^), and 3.08 ppm (**L^2^**–Hg^2+^)); water peaks could not be assigned unambiguously in **L^3^**–Hg^2+^ due to the complex NMR splitting pattern. MALDI mass spectra were recorded with a *p*-nitroaniline matrix (10 mg/mL in CH_3_CN) on a Bruker UltraFlex III TOF/TOF mass spectrometer in positive-linear mode using an Nd:YAG laser (λ = 266 nm). IR spectra were recorded in KBr on a Bruker Vector 22 spectrometer.

### Synthesis of Thiacrown-Ether L^3^

A dropwise addition of diethyl azodicarboxylate (2.5 eq.) to the suspension of thiacalix[4]arene **L^1^** (1 eq.) with PPh_3_ (2.5 eq.) and C_12_H_25_OH (10 eq.) in toluene at room temperature followed by stirring for 8 h at 40 °C, distillation of solvent under reduced pressure, and washing of the oily residue with MeOH afforded the intermediate **L’^1^** as previously reported [17]. The intermediate (1.21 g, 1.11 mmol, 1 eq.), PPh_3_ (1.74 g, 6.64 mmol, 6 eq.), and 3,6-dithia-1,8-octanediol (1.01 g, 5.54 mmol, 5 eq.) were then suspended in 30 mL toluene, and diethyl azodicarboxylate (1.16 g, 6.64 mmol, 6 eq.) was added dropwise at 0 °C. The suspension was stirred for 8 h at 20 °C and then for 8 h at 40 °C. The solvent was evaporated and methanol (40 mL) was added to the residue; the precipitate was filtered and purified by column chromatography (hexane:ethylacetate = 4:1).

Compound **L^3^**. Yield 0.28 g (21%). T_m_ = 186 °C. R_f_ (*n*-C_6_H_14_:EtOAc = 6:1) 0.54. ^1^H NMR (CDCl_3_): 7.41 (s, 4H, H_12_), 7.35 (s, 4H, H_3_), 4.09 (m, 4H, H_7_), 3.76 (t, 4H, *J* 8.4, H_16_), 2.60 (m, 4H, H_8_), 1.74 (s, 4H, H_9_), 1.31 (s, 18H, H_6_), 1.29 (s, 18H, H_15_), 1.28–1.20 (m, 40H, H_17–26_), 0.90 (t, 6H, *J* 6.8, H_27_). ^13^C NMR (CDCl_3_): 158.3 (C_1i_), 156.3 (C_10i_), 146.0 (C_4i_), 145.5 (C_13i_), 129.5 (C_3_), 129.3 (C_2i_), 128.3 (C_11i_), 127.4 (C_12_), 72.4 (C_7_), 68.6 (C_16_), 34.6 (C_5i_), 34.4 (C_14i_), 33.4 (C_8_), 32.7 (C_9_), 32.1 (C_17_), 31.7 (C_15_), 31.5 (C_6_), 30.3 (C_18_), 29.8 (C_19–22_), 29.5 (C_23_), 28.3 (C_24_), 26.0 (C_25_), 22.8 (C_26_), 14.3 (C_27_). *m/z* (MALDI) (%) 1193.9 (100) [M – S + Na]^+^. Anal. calcd. for C_70_H_106_O_4_S_6_: % C 69.83, H 8.87; found: C 69.68; H 8.90. IR (KBr, $\tilde{\nu }$/cm^–1^) 2924 (C–H), 1438 (C_Ar_–C_Ar_), 1263 (C_Ar_–O–C_Alk_), 1086 (C_Alk_–O–C_Alk_).

Extraction of metal picrates from water into CH_2_Cl_2_ followed a typical procedure and was conducted in triplicate to ensure reproducibility. Picric acid, Hg(ClO_4_)_2_, and other metal (Ag, Cu, Pb, Hg, Zn, Cd) nitrates were dissolved in 3 mL of H_2_O to the final concentration *c*_HPic_ = 2 × 10^−4^ M and *c*_M_ = 10^−2^ M. Ligands **L^2^**–**L^4^** were dissolved in 3 mL of CH_2_Cl_2_ (*c* = 4 × 10^−4^ M). The 1:1 CH_2_Cl_2_–water system was stirred for 30 min and then left for 15 min for phase separation. UV/Vis spectra of the aqueous phase were recorded before and after extraction, and maximum absorbances at 355 nm (ε = 1.45 × 10^4^ M^–1^ cm^–1^) (*A*_0_ and *A*_i_, respectively) were measured. The extraction percentage (%*E*) was calculated as follows:$$\%E=\frac{A_{0}-A_{i}}{A_{0}}\times 100.$$

The extraction constants were calculated as follows:$$\log K_{\mathrm{ex}}^{\mathrm{Ag}}=\log D_{\mathrm{Ag}}-n\log{\left[L\right]}_{\mathrm{org}}-\log{\left[{\mathrm{Ag}}^{+}\right]}_{\mathrm{aq}},\;\log K_{\mathrm{ex}}^{\mathrm{Hg}}=\log(D_{\mathrm{Hg}}/\left(1-\alpha \right))-n\log{\left[L\right]}_{\mathrm{org}}-2\log{\left[{\mathrm{Hg}}^{2+}\right]}_{\mathrm{aq}},$$
where *D* is the distribution ratio, *D* = α/(1–α), α is the degree of extraction (α = %*E*/100), *n* is the ligand-to-metal ratio in the complex, [L]_org_ is the equilibrium concentration of the ligand in the organic phase, and [Ag^+^]_aq_ and [Hg^2+^]_aq_ are equilibrium concentrations of the metal ions in the aqueous phase.

DLS measurements were performed on a Zetasizer Nano particle size analyzer (Malvern Panalytical, Malvern, UK) in PCS1115 glass cuvettes that were thermostated at 20 °C. Three independent experiments were recorded for each sample; data were processed by the Malvern DTS program.

UV/Vis spectra were recorded in quartz cuvettes (*l* = 1 cm) on an AvaSpec-2048 spectrophotometer in the wavelength range of 200–500 nm with a resolution of 1 nm. Standard spectrophotometric titration of the compounds **L^2^** and **L^3^** was performed with 0.1–10 eq. of metal salts. To minimize the dilution effects, microvolumes of 10^–3^ M salt solutions were added to 1 × 10^−5^ M solutions of ligands. To avoid solvent evaporation, the temperature of the qpod 2e Peltier-controlled cuvette holder was maintained at 20.0(1) °C. Correction for dilution and baseline correction were applied to the spectra upon data processing.

Monolayers of the ligands **L^2^**–**L^4^** were formed using the Langmuir method by spreading their CHCl_3_ solutions (*c* = 5 × 10^−5^ M, *V* = 35–105 μL) onto the water subphase with a microsyringe at the air–water interface on a KSV NIMA Teflon trough equipped with a Pt Wilhelmy plate and two polyacetal barriers. The concentration of CHCl_3_ solutions was controlled spectrophotometrically before spreading onto the water subphase. SPOT was monitored (with an accuracy of ±2 mV) using a KSV SPOT1 instrument with a vibrating plate electrode at a distance of ca. 1 mm from the water subphase and a stainless-steel counter electrode immersed in water.

Following monolayer production conditions were employed: the time of spreading solvent evaporation was 15 min, monolayer compression rate was 5 cm^2^/min, and *t* = 25 °C. The KSV Nima/Attension 2.2 and Origin programs were used for processing surface pressure/SPOT–mean molecular area data. Monolayers were vertically deposited onto quartz substrate, which was degreased with EtOH and washed with water before use. The 10^−3^ M solutions of metal salts in the water subphase were used for complexation study.

For XPS measurements, compound **L^3^** was transferred from air–Hg(ClO_4_)_2_ aqueous solution interface onto quartz substrate using the LB technique at a transfer pressure of 20 mN/m. The surface of the so-modified quartz substrate and bare quartz (which was immersed into 10^−4^ M Hg(ClO_4_)_2_ and, then, dried in the air for 24 h) were analyzed using an Omicron spectrometer (UK) with monochromatic Al Kα radiation in the fixed analyzer transmission mode. The analyzer pass energy was set at 100 eV for registration of survey and 40 eV for measuring high-resolution spectra. The spectra were measured with a step size of 0.1 eV at room temperature. The energy scale of the spectrometer was calibrated according to the standard procedure, taking into account the following binding energies: 932.7, 368.3, and 84.0 eV for Cu 2p_3/2_, Ag 3d_5/2_, and Au 4f_7/2_, respectively. The residual pressure inside the analysis chamber was lower than 1.0 × 10^−9^ mbar. Photoelectron spectra were approximated by a sum of Gaussian functions after Shirley-type background subtraction. Sample charging was corrected by referencing the Si 2p_3/2_ peak of Si^0^ state with a binding energy of 99.34 eV [35].

Computations of compound **L^2^** and **L^2^**–Ag and **L^2^**–Hg complexes were performed using the Gaussian 16 suite of programs [36] and def-TZVP basis set (TZVP stands for valence triple-zeta polarization) [37]. The ground-state structures were optimized at DFT using the Becke 3-parameter Lee–Yang–Parr (B3LYP) function [38,39]. The D3 London dispersion correction was applied as implemented in Gaussian [40]. The electronic spectra were simulated using time-dependent DFT [41,42,43] by calculating the first 50 vertical excitations from the ground state (S_0_) equilibrium geometries with cam-B3LYP long-range-corrected functional, which yield good results for charge-transfer systems [44,45,46]. Vertical transitions were broadened with the Gaussian function of full width at half maximum of 0.24 eV. The dipole length formalism was used to calculate the oscillator strengths. The excitation energies were consistently red-shifted by 0.25 eV upon cam-B3LYP calculations in order to better match experimental spectral curves.

The single-crystal X-ray diffraction data for [Hg_2_**L^4^**Cl_4_], [Ag_2_**L^2^**_2_](ClO_4_)_2_, and [Ag_2_**L^2^**(ClO_4_)_2_]*_n_* were collected on the ‘Belok’ beamline of Kurchatov Synchrotron Radiation Source (National Research Center ‘Kurchatov Institute’) using a Rayonix SX165 CCD detector at *λ* = 0.7450 (for [Ag_2_**L^2^**_2_](ClO_4_)_2_ and [Ag_2_**L^2^**(ClO_4_)_2_]*_n_*) and 0.80246 Å (for [Hg_2_**L^4^**Cl_4_]). The frames were collected using an oscillation range of 1.0° and φ scan mode. The data were indexed and integrated using the iMOSFLM utility from the CCP4 program suite [47] and then scaled and corrected for absorption using the Scala program [48]. Structures were solved using Olex2 software [49] by direct methods with SHELXT [50] refined by the full-matrix least-squares on *F*^2^ using SHELXL [51]. Non-hydrogen atoms were refined anisotropically.

Crystallographic Data Centre at www.ccdc.cam.ac.uk/datarequest/cif (accessed on 14 August 2021) (CCDC 2103219 ([Hg_2_**L^4^**Cl_4_]), 2103218 ([Ag_2_**L^2^**_2_](ClO_4_)_2_), and 2103220 ([Ag_2_**L^2^**(ClO_4_)_2_]*_n_*)).

In a crystallization tube (4 mm diameter, 15 cm height), a solution of the compound **L^2^** or **L^4^** (4.6 μmol) in CHCl_3_ (1 mL) was layered with a 1/1 CHCl_3_/MeOH mixture (1 mL). A solution of AgClO_4_ (4.6 μmol for [Ag_2_**L^2^**_2_](ClO_4_)_2_ or 46 μmol for [Ag_2_**L^2^**(ClO_4_)_2_]*_n_*) or HgCl_2_ (9.2 μmol) in MeOH (1 mL) was carefully added. Slow diffusion at 25 °C over 7–14 days produced colorless crystals suitable for X-ray diffraction studies.

[Hg_2_**L^4^**Cl_4_]: Found: C, 48.6; H, 5.05. C_92_H_116_Cl_4_Hg_2_O_8_S_12_ requires C, 48.5; H, 5.1%.

[Ag_2_**L^2^**_2_](ClO_4_)_2_: Found: C, 52.9; H, 5.7. C_92_H_120_Ag_2_O_8_S_12_*2(ClO_4_) requires C, 52.9; H, 5.8%.

[Ag_2_**L^2^**(ClO_4_)_2_]*_n_*: Found: C, 37.9; H, 4.1. C_46_H_60_Ag_2_Cl_2_O_12_S_6_*2CHCl_3_ requires C, 37.9; H, 4.1%.

## 4. Conclusions

A comparative study of the interaction of thioethers of different topologies (linear bis-(methylthio)ethoxy derivative **L^2^**, O_2_S_2_-thiacrown-ether **L^3^**, and O_2_S_2_-bridged thiacalix[4]tube **L^4^**), attached to the thiacalixarene scaffold, with metal cations allowed us to draw the following conclusions. For all compounds **L^2^**–**L^4^** highest liquid-phase extraction selectivity was revealed for Ag^+^ and Hg^2+^ ions vs. other soft metal ions. In linear thioether **L^2^** and thiacalixtube **L^4^**, binding of Ag^+^ and Hg^2+^ was evidenced by a blue shift of the band at 303 nm and the appearance of a charge-transfer band centered at 330 nm, respectively. Comparison of experimental data obtained by several modern methods (UV/Vis and NMR spectroscopy and X-ray diffractometry) with the results of quantum-chemical calculations of UV/Vis spectra of Ag and Hg complexes of the thioethers **L^2^** and **L^4^** showed that coordination around the Hg^2+^ center involving alkyl sulfide atoms and coordination around the Ag^+^ center involving aryl and alkyl sulfide atoms take place. It was also proved that for compound **L^3^**, alkyl sulfide groups participate in the binding of both Hg^2+^ and Ag^+^ ions. In spite of the low hydrophilicity of the ligands **L^2^**–**L^4^**, reproducible Langmuir monolayers are formed at the air–water interface, which are able to adsorb Hg salts from the water subphase. These results demonstrate the potential of sulfur functionalities on a calixarene scaffold as a receptor fragment towards Hg^2+^ ions, which could be useful in the development of Hg^2+^-selective water purification systems. Future work will focus on the uptake capacity and sensing of Hg^2+^ ions by the thioether ligands on a calixarene scaffold in ultrathin films.

## Data Availability

The data presented in this study are available in article and Appendix A.

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
