# Peer review of "Thiacalixarenes with Sulfur Functionalities at Lower Rim: Heavy Metal Ion Binding in Solution and 2D-Confined Space"

_ijms, 2022, doi:10.3390/ijms23042341_

Round 1

Reviewer 1 Report

This paper describes the selective complexations of thiacalixarenes modified by sulfur containing substituents with Ag+ and Hg2+ in solution and on the Langmuir−Blodgett films. The results are supported by UV-vis and NMR spectroscopies, single crystal X-ray diffractions, XPS as well as the theoretical calculations. This is a carefully done study and the findings are interesting. Thus, this paper is worth publishing in International Journal of Molecular Sciences with minor revisions. Some additional comments are listed below.

1) Page 4, line 110 − Page 5, line 149, page 14, line 468 − page 15, line 469 and Figure S3: The “lg” is not correct. They should be revised to “log”.

2) Figure 3−5: These figures are difficult to understand. Authors should magnify them. I recommend they are exchanged new ones changed direction slightly to show bonds around metal ions more clearly.

Author Response

The authors thank the reviewer for the comments to the manuscript. The "lg" was corrected into "log" and Figures 3-5 were enlarged to to fit two-column width. A clearer representation of the bonds around metal ions is provided in Figure S5a-c.

Reviewer 2 Report

Manuscript

The manuscript entitled “Thiacalixarenes with Sulfur Functionalities at Lower Rim: Heavy Metal Ion Binding in Solution and 2D-Confined Space” by Anton Muravev and co-workers reports the interface receptor features of  three topologically different thiacalixarene thioethers L2, L3 and L4  towards metals in bulk liquid phase and Langmuir monolayers. The authors evaluated how the thioether topology affects the metal ion binding mechanisms and selectivity.

The subject is interesting and is on line with the interest of the group on developing thiacalixarenes with improved sensing  features. The results are supported by recurring to different experimental techniques and theoretical calculations.

However, some reformulations are required in order to show the real novelty of the achievements when compared with the previous results of the authors.

So, the authors must discuss and compare the results obtained with  the previous ones and to show how the new design improves or not the sensing features of the thiacalixarenes with sulfur functionalities.

Other aspects that the authors must consider in a revised version:

  1. The synthetic approach represented in Scheme 1 must be improved since it is necessary to read previous articles of the authors on the subject in order to understand the methodology used. The authors must also try to improve the structures of the thiacalixarenes.
  2. In  section Synthesis and Characterization of Ligands, the authors must restrain the designation of the ligands just to one form (L2, L3 and L4). How the scaffold L1 was obtained?
  3. The authors must clarify why in compound L3 the 1,3-alternate stereoisomeric form was confirmed by closely spaced NMR signals of aryl protons at 7.41 and 7.35 ppm and tert-butyl protons at 1.31 and 1.29 ppm.
  4. Line 114 - indicate the meaning of D in lgDAg and of alfa since the experimental part only appears posteriorly.
  5. Line 133 - the authors refer that no temporally stable aggregates in case of thiacrown L3, whereas stable spherical aggregates were detected after extraction of Hg2+ (d = 437±12 nm) and Ag+ ions (d = 347±12 nm) by the thioether L2 (Figure 2b).  What happen with L4?
  6. If possible in Figure 5 the representation of the crystal packing must be improved.
  7. Line 255 in Figure 7 Indicate the colour of each atom.
  8. Concerning the synthesis of the new,  thiacrown-ether L3 all the experimental  details must be reported  starting  from thiacrown-ether L1.
  9. Line 458 concerning the Extraction of metal picrates from water into CH2Cl2 the volumes used must be indicated. Define Ao, Ai etc and all the variables in the equations presented.
  10.  Other references that must be considered  https://doi.org/10.3390/molecules25030612. 

Author Response

The authors thank the reviewer for the comments. The results obtained were compared with  the previous ones and and it was discussed how the new design improves or not the sensing features of the thiacalixarenes with sulfur functionalities (lines 445-455).

A point-by-point response is given below:

1) Scheme 1 was redrawn to show the structure of the intermediate towards ligand L3. A detailed explanation of the synthetic approach was added to the Results section (lines 85-96).

2) Designation of the ligands was standardized into one form. The ligand L1 was prepared by the base-mediated reaction of 4-tert-butylphenol with elemental sulfur in tetraglyme, the reference reporting its synthesis was added (line 86).

3) Explanation of the existence of compound L3 in 1,3-alternate stereoisomeric form was added (lines 106-116).

4) This was added to the text (lines 134-138).

5) L4 formed temporally stable nanosized aggregates only with Ag+ ions (lines 149-153).

6) Crystal packing with shaded carbon atoms providing a clearer view of the metal coordination environment was added to Figure 5 and was enlarged up to two-column width.

7) These were added to Figure 7.

8) More detailed experimental data for the ligand L3 was provided (lines 479-487).

9) These were added to the main text (line 501, 502). Ao and Ai were defined in lines 505, 506.

10) This reference was added to line 57 as ref. 6.

Reviewer 3 Report

The authors report a comparative study of the interaction of thiacalixarene thioethers with metal cations. The paper is well written and the reported results have relevance in supramolecular chemistry and in water purification field. I believe that the manuscript can be accepted for publication in this Journal after minor revisions.

In the section 2.1 (Synthesis and Characterization of Ligands), the authors discuss about compounds named 1, 2, 3 and 4; however, in scheme 1, compounds are named as L1, L2, L3, and L4. Compounds must be named in the same way.

In the Conclusions section the future directions of the proposed research should be addressed.

Author Response

The authors thank the reviewer for comments to the manuscript. The compounds named as 1, 2, 3, and 4 were replaced by L1, L2, L3, and L4. In the end of Conclusions section, following sentence was added "Future work will be focused on the uptake capacity and sensing of Hg2+ ions by the thioether ligands on a calixarene scaffold in ultrathin films" to address the future directions of the proposed research.

Reviewer 4 Report

The work seems adequate, although I would recommend that more emphasis should be placed  on the final aim of the obtained compounds and their relevance both for research purposes but also on practical utility. Also, some of the references seem outdated and I have some doubts about the necessity of some many authors for what seems like a very basic research project.  

Author Response

We thank the reviewer for the comments. More emphasis was placed on the aim of the compounds and their relevance for research (lines 80-82) and practical application (lines 450-455). The reference list was updated with more recent works (refs. 6, 19). The author list is large due to the large number of methods performed by different people: titration experiments (1 co-author), theoretical calculations (1 co-author), X-ray diffractometry (5 co-authors), monolayer measurements (2 co-authors), XPS study (1 co-author), and 3 other co-authors were involved in review and editing of the manuscript text.

Round 2

Reviewer 2 Report

The authors improved the manuscript according with my comments and  is now in conditions to be accepted.